# Deep Meaning in Scenic Assessment: Seeing around the Bend

## Patrick Alan Miller

Landscape Architecture Program, College of Architecture, Arts, and Design, Virginia Tech, Blacksburg, VA 24061, USA; pmiller@vt.edu; Tel.: +1-540-577-7299

**Abstract:** Many of today's scenic assessment tools were developed initially to preserve the visual quality of public lands and to mitigate the negative visual impacts of large-scale landscape alterations, such as timber harvesting, mineral extraction, and renewable energy production. However, we are beginning to see more interest today in preserving scenic views on non-public lands. This essay makes a case for additional scenic assessment tools that reveal deep meaning in the landscape. Deep meaning is different than the immediately visible attributes of the landscape. It includes things that come to mind when looking at the landscape and are shared by people familiar with that landscape. Understanding the concept of deep meaning can be difficult. The author describes how deep meaning in the landscape was first revealed to him. Two non-landscape examples are then used to demonstrate different aspects of deep meaning. First, the rocks in a rock garden show the personal nature and attachment of deep meaning. Second, the wording on two wine bottle labels illustrates the distinction between surface meaning and deep meaning. Poetry is then examined as a means of conveying deeper landscape meaning. Lastly, four different landscape contents categories that are used in the proposed Virginia Scenic Viewshed Program demonstrate how deep meaning content can be used in the scenic viewshed assessment. The conclusion is that deep meaning would be a valuable addition to the scenic assessment of non-public land viewsheds, particularly if the assessment process involves the public.

**Keywords:** scenic quality/assessment; viewshed; deep meaning; visual assessment/quality; scenic landscape

## 1. Introduction

People are becoming more concerned about maintaining the scenic quality of their nearby landscapes, those landscapes that they often experience. Two things are driving this concern. First, is the slow, but noticeable, deterioration of scenic quality in many landscapes due to incremental development. Incremental development takes place slowly and seems acceptable as it is occurring, but the cumulative effect is often the destruction of scenic quality. Second, having been locked down during the pandemic, people are now more interested in seeing and experiencing scenic landscapes, particularly those close at hand. This desire is one reason for the increase in the sale of recreational vehicles in the U.S. [1]. For visual management professionals, this increased interest in the scenic quality of landscapes is exciting, but are we ready?

Many of the scenic quality tools we have today were initially developed to preserve the visual quality of public lands and to mitigate the negative visual impacts of large-scale landscape alterations such as timber harvesting, mineral extraction, and renewable energy production [2]. However, are these tools suitable for saving Aunt Betsy's favorite views? Hers are views that are closer to her home and seen more often. They are landscapes that have more to do with personal meaning and attachment to people living in a certain region than managing natural resources. Do we need some additional tools that will allow us to capture the more personal experience that occurs when someone views a landscape they love? In this essay, I am calling this the "deep meaning" that can occur when viewing the landscape. What is meant by deep meaning will be explained further below. This

article is not a research paper, it is an essay. The purpose of this essay is to describe the phenomenon that I am calling deep meaning and to make the argument that additional visual management tools are needed as we move toward scenic management that is more citizen-oriented at the regional level. This is not meant to be a critique of visual management tools that are working well today in the context of public lands at the federal level, except to the extent that they may limit our openness to other tools in a different context.

## 2. Background and Context

To understand the need for deep meaning in scenic landscapes, the evolution and purpose of some of the existing scenic management tools will be described. Next, a personal experience that made me realize that something was missing in our visual assessment tool kit will be shared. The missing tool is the ability to capture deeper, more personal meanings conveyed by the visual experience. Understanding what is meant by deep meaning in the landscape can be complex. So, two non-landscape examples of deep meaning will demonstrate the personal nature of deep meaning and how it can be conveyed. In the first, a rock garden reveals the personal nature and attachment conveyed by deep meaning. In the second, wine bottles and labels reveal how deep and surface meanings can be conveyed, and the importance of deep meaning in addition to quantitative measures in assessing the quality of a wine. Next, the power of poetry to convey deep meaning in the landscape will be examined. Finally, I examine how deep meaning was serendipitously discovered and applied in the development of scenic assessment criteria for landscapes in the Commonwealth of Virginia. These criteria along with more traditional assessment tools are being proposed for the new Virginia Scenic Viewshed Program.

### 2.1. Seeing Back: Context

During the 1960s and 1970s, great work was conducted in the development of methods and tools for managing the scenic landscape. As mentioned above, much of this work was carried out to plan and manage large-scale public landscapes. This period is often referred to as the Environmental Movement [3]. The U.S. Congress enacted many new laws during this period. These included laws such as the Land-Use Policy Act, Multiple-Use Sustained-Yield Act of 1960 [4], and The Environmental Policy Act of 1969 [5]. While these acts did not directly address scenic or visual quality, they provided a multiple-use context under which scenic management could be included in the planning and management of public lands.

Those early landscape architects did a fantastic job. They built on the ideas and theories of earlier landscape architects such as Dame Sylvia Crowe in Great Britain, author of The Landscape of Forests and Woods [6]; Burt Litton, author of Forest Landscape Description and Inventories [7] and Landscape Control Points [8]; Donald Appleyard, one of the authors of View from the Road [9]; and Kevin Lynch, author of several books including Image of the City [10]. While there was a blossoming of new visual management ideas, concepts, and theories, there was still a predominant belief by many that scenic beauty was in the eye of the beholder. This often came from a positivistic or rationalistic view of the world. This philosophy favors a bipolar, or an objective-subjective conception of the world. This is a view that favors analytic reason over experiential understanding. Many with this view did not believe that scenic quality could be assessed or evaluated. Those early landscape architects had to sit around the table with other resource managers such as foresters, geologists, and range conservationists, all of whom had metrics for the resources they were managing. Some landscape architects felt that words such as scenery or scenic quality fed into the belief that the visual aspect of the landscape was too subjective and was not suitable to use in planning and management of the landscape. These early landscape architects wanted the scenic character of the landscape to be a resource to be managed, just like timber, minerals, and range resources. To gain credibility with other resource management professionals, they came up with the name "visual resource management." They also developed metrics for scenic assessment that could be measured more objectively, such as

visual complexity/diversity. Visual assessment procedures were adopted by many federal land management agencies [11]. These early landscape architects need to be commended for successfully obtaining credibility for the planning and management of the visual landscape. However, maybe they were too successful? Why has there not been more scenic assessment theory and methodology development since then? In his systematic examination of the evolution of visual impact assessment, Smardon notes, "After the 1980's there has been little new work in visual resources assessment methodologies aside from developing visual simulation digital technology" [11]. So, this may account for what seems like a gap in the visual assessment literature as we move toward a theory such as deep meaning. As long as visual assessments are being conducted in the context of large-scale, federally owned lands with an emphasis on objective measures, there was no perceived need to move in the direction of deep meaning. However, the context is changing.

In the United States, we see increased interest in scenic assessment and preservation on non-public lands in places such as Virginia [12] and Tennessee [13]. Both Virginia and Tennessee are proposing scenic viewshed registers that include a public nomination or participation process. The engagement of the public is critical in gaining support for a state scenic viewshed register, such as the one being proposed in Virginia. Europe is already engaging the public in the implementation of the European Landscape Convention [14]. A recent study argued that, " . . . a method seeking to fulfill both sustainability and a public participation agenda would have to combine quantitative forms of socio-economical assessment with a qualitative measurement of the cultural appreciation of landscapes" [15]. In a study comparing the citizen engagement processes used in three autonomous Spanish communities, it was concluded that the best community engagement process included citizens actually experiencing the landscape. This suggests the need for more experience-based assessment tools such as deep meaning [15].

While there may not have been much movement in the visual assessment literature since the 1980s, there was movement in place theory. There is a plethora of literature on place theory, much of it exploring the nature of the phenomenon or the relationship between people and their environment. Sense of place theory and place attachment [16] provide a broader theoretical bases for what is being advocated in this essay. It is reassuring to know that there are potential aesthetic benefits from the experiential relationship between people and the natural environment [17]. However, there is still a gap between theory and application. As Lee and Matarrita-Cascante point out, "The notion of place has been discussed within many different fields. Nonetheless, not many conversations are happening between the different fields that study this concept. This has resulted in the creation of a body of knowledge that, on instances, tend to be secluded from the knowledge created in other fields" [18]. It is the purpose of this essay to advocate an application of some of the experiential place theory in a form that can be applied in the landscape planning realm of scenic landscape assessment. This application is a more limited form of sense of place theory or experiential understanding and is being referred to here as "deep meaning." This essay is not advocating that deep meaning be a replacement for existing approaches to scenic assessment but rather as an addition to those traditional methods in places where citizen involvement and support are important. Deep meaning is complex and difficult to understand. This essay provides several examples of deep meaning, including two non-landscape examples, in order to convey the personal nature of deep meaning and how words can help people envisage deep meaning. An example of the how deep meaning can be used of in visual assessment is also provided.

### 2.2. What Is "Deep Meaning?"—Learning to See the Unseeable

When we view something, our mind does two things. First, our mind creates a visual composition or image of the physical elements being viewed. These physical elements can be described and are called surface meanings. The second thing our mind does is try to understand what we see by referencing content or memories stored in our minds. These memories or content are "deep meaning." Our minds do this automatically because our

minds are trying to make sense of what we are looking at. Often this happens without conscious awareness. Sometimes, we have to look a little closer or reflect a bit on the image to obtain its deep meaning. Words, particularly those of a poetic nature, can stimulate this mental process. Deep meaning is more personal and often more powerful or evocative than surface meaning. These meanings may differ between individuals but are often shared or similar to the meanings attributed by others. This is following a more existential view or philosophy. In this philosophic position, the world is not bi-polar, it is neither objective or subjective. Rather, it is a continuum, a view of the world based on experiential understanding. Shared meanings of a viewshed would be important when involving people in the preservation of scenic viewsheds at the local level. The following section will examine some examples of deep meaning and the implications for scenic landscape assessment.

*2.3. Learning to See?—Discovering Deep Meaning*

In 1977, I worked for the BLM (Bureau of Land Management) in New Mexico. The majority of my work was in recreation planning and design. However, this was also when the BLM visual resource management procedures were being implemented in different BLM districts. Being the only landscape architect in the office, my supervisors asked me to complete some visual contrast ratings on recently completed range improvements, such as wells, stock tanks, and fences. The procedure assesses how much visual contrast will exist between the form, line, color, and texture of the proposed alteration and that of the existing landscape. Normally, contrast ratings should be completed before the construction, using visual simulations of the proposed alteration. The procedure is described in BLM Manual 8431—Visual Resource Contrast Rating [19]. However, since the procedure was just being implemented and my supervisors were curious about the degree of visual contrast and possible mitigating measures that might apply to future projects, they decided to perform some contrast ratings on the recently completed range improvements.

The basic philosophy underlying the BLM Visual contrast rating system is stated as: the extent to which an alteration affects the visual quality of a landscape depends on the visual contrast created between a project and the existing landscape. The contrast can be measured by comparing the project features with the major features in the existing landscape. The basic design elements of form, line, color, and texture are used to make this comparison of the visual contrast created by the project with the natural landscape. This assessment process provides a means for determining visual impacts and for identifying measures to mitigate visual impacts.

For the most part, the contrast rating procedure seemed to work as intended for the range improvements. However, there was one improvement that did not feel right. It was the replacement of a windmill pump (Figure 1) with an electric well pump (Figure 2). Indeed, the electric pump was smaller and had less visual contrast with the natural landscape's form, line, color, and texture than the high-in-the-air windmill.

However, it was clear that when I looked at the windmill, I saw more than its form, line, color, and texture. It conveyed something deeper. The windmill evoked images in my mind of the old west—images of cattle, cowboys, homesteaders, and the timeless turning of the windmill blades. When we view something, we see more than the visual attributes of the physical object. The image activates networks of neurons in our brain that convey meanings about the object, deep meaning. The fact that the BLM visual contrast rating procedure did not capture the deep meaning of the windmill has bothered me for more than 40 years. Where does deep meaning come from, and how do we acquire it?

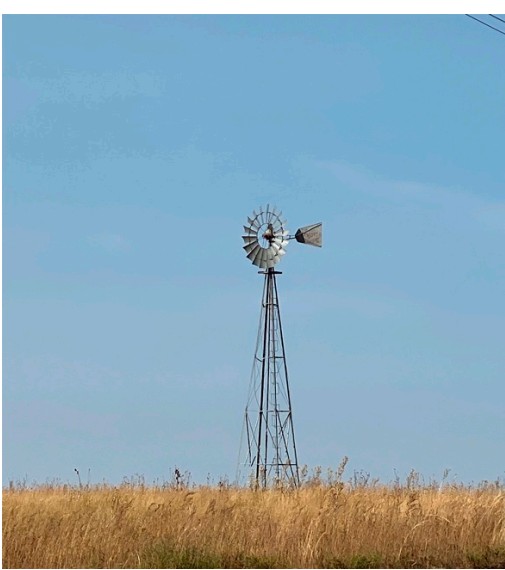

**Figure 1.** The Windmill well pump has greater visual contrast but also deep meaning related to cattle country of the West. (Photo by author).

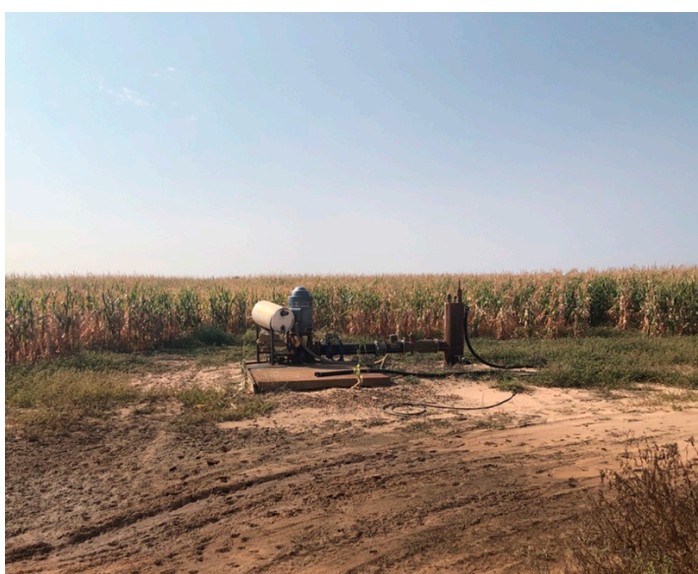

**Figure 2.** The electric well pump is smaller and has less visual contrast than a windmill, but it lacks deep meaning and does not seem like it belongs (Photo by author).

Deep meaning in the landscape is complex and often difficult to understand. In order to help clarify what is meant by deep meaning, two non-landscape examples of deep meaning will be described.

*2.4. Aunt Georgie's Rock Garden: Seeing the Personal Nature of Deep Meaning*

I had a lovely aunt who lived in a small town in Colorado. Her name was Georgie. Aunt Georgie knew that I was a landscape architect. Whenever I visited her, she would insist that I go into her backyard and look at her rock garden. Her backyard was rather nondescript. It was a fenced area with grass edged by planting beds and a tree in the middle. The rocks were displayed in the planting beds. She would walk me along the beds picking up rocks as we went. Each rock had a story. Some looked like turtles, rabbits, or other small animals. Some of the rocks were souvenirs from trips she had taken with her husband, my Uncle Oscar. She would tell me where they went, why they went there and why she chose that particular rock as a souvenir of the trip. Some rocks were just optically

fascinating, containing sparkles, colors or shapes that were visually interesting. I could tell that Aunt Georgie experienced something very personal, a deep meaning, that was very important to her for each rock. She cared for and valued each rock. Rocks are like miniature landscapes, particularly to those who have experienced a landscape at multiple times in their life.

Aunt Georgie's rocks are like Aunt Betsy's landscapes; deep meaning makes them personal and memorable. Deep meaning is essential when undertaking scenic assessment at the local level. How can we capture deep meaning in the landscape when that meaning may not be immediately evident? Obviously, every deep meaning felt by every individual who views a particular landscape cannot be captured. However, is there enough in common for a group who occupy the same geographic area and have experienced a landscape many times to have similarities in their experience? This essay is advocating that there is.

### 2.5. Seeing Deep Meaning in the Evaluaiton of Wine: Words That Describe the Deep and Shallow Meaning

Detail and description are essential to convey a deep meaning. Deep meaning in the landscape, however, can be elusive. A deeper, more profound sense of beauty and mystery is easily overlooked in the practical world of getting things done. Wine is a good analogy for understanding how deep meaning can be conveyed. At the most superficial level, wine is fermented fruit juice. Most wines have descriptive or surface information, such as the type of grape, the year it was vinted, the name of the vintner, and the location where grapes were grown. Most people want to know this when they purchase a bottle of wine (Figure 3). This is surface meaning and easy to convey and understand.

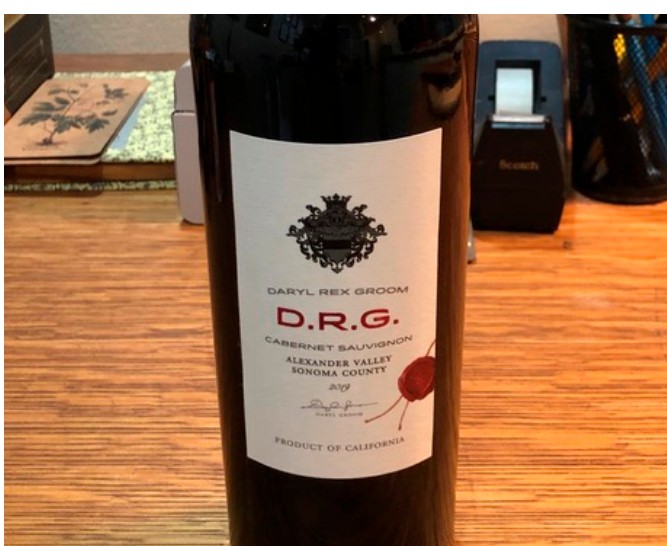

**Figure 3.** Bottle of wine with descriptive, factual information including the type of grape, where the grapes were grown, the year vinted and the name of the vintner. These attributes are visually evident on the label and convey surface meaning; essential facts about the wine (Photo by author).

However, we know that great wines also have complex flavors and fragrances. Drawing air across one's tongue can enhance flavors and unique smells that change as one sips the wine. Describing them is more complicated. The same type of wine can have subtle differences in flavor that cause them to be more desirable. More poetic adjectives are often used to describe and name these wines. Words such as "smoky flavors" or "slightly bitter finish" are used to describe flavor variations. Sometimes, the words may even seem contradictory. Is a smokey flavor supposed to make a wine taste better? The use of these types of words causes the reader to stop and think, thus revealing subtle flavors conveyed as deep meaning.

The name of the wine on the label of the bottle depicted in Figure 4 is named, Fog & Light. The title conveys something more mysterious than just the type of grape or where it was grown. The words fog and light seem somewhat contradictory. How can both fog and light occur at the same time? Perhaps, it conveys a sense that the flavors of the wine would be fragrant and subtle while simultaneously providing new flavors. The contradictory words are poetic and intrigue the potential buyer, making them curious about the subtle flavors in a wine, similar to deep meaning in the experience of a landscape. In addition, the bottle has a wax seal over the cork—conveying that special attention and extra care was given to this bottle of wine. The label also includes surface information about the type of grape, the vintner, and where it was grown. However, one would expect that this name and its wax seal would convey deep meaning and that the deep meaning would be provocative and enticing to people. Thus, people will be more likely to buy the wine.

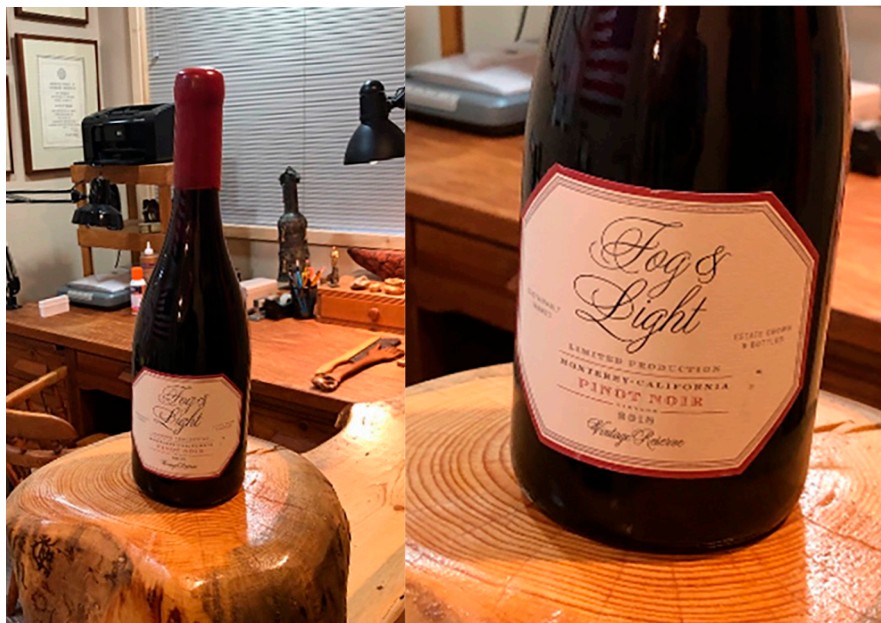

**Figure 4.** The image on the left is a bottle with a wax seal on the cork and a closeup of the label with the name Fog and Light. The wax seal conveys extra care for a better wine and the poetic name Fog & Light conveys deep meaning, much like the flavors of good wine (Photos by author).

While it is often necessary to provide surface or visually descriptive information about a viewshed, for example size and location, the use of adjectives that provide deep meaning would be more evocative and probably attract more viewers. Involving local people in naming and describing viewsheds would be particularly important at the local level. Again, that personal meaning would help encourage people to preserve their valuable scenic viewsheds. Can poetry be used to convey deep meaning?

### 2.6. Descriptive Deep Meaning in the Landscape: Williams Wordsworth and Poetry

The well-known early romantic era poet, Williams Wordsworth, was the first to describe the landscape using deep meaning, although he did not call it "deep meaning." Instead, it was his poetry that provided deep meaning. He wrote a guidebook titled "A Guide to the Lake District of England." It was published anonymously in 1810, and a second edition was published under his name in 1820. The Lake District of England was very scenic (Figure 5). There was speculation about why he published the guidebook. Some say he needed the money. He said, "What I hope to accomplish was to give a model of the manner in which to topographical descriptions ought to be executed, in order to there being either useful or intelligible, by evolving truly and distinctive one appearance from another" [20].

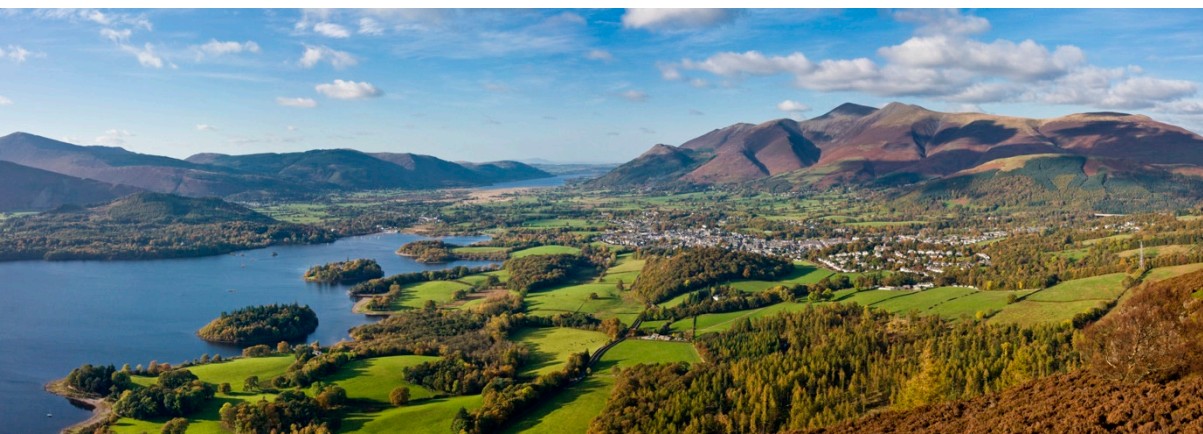

**Figure 5.** The Keswick Panorama in the scenic Lake District of Cumbria, England. (Photo by David Diliff from Creative Commons). No changes to were made to the photo. Available on line https://upload.wikimedia.org/wikipedia/commons/thumb/a/ab/Keswick_Panorama_-_Oct_ 2009.jpg/1280px-Keswick_Panorama_-_Oct_2009.jpg (accessed on 22 July 2022). [21]. License: CC BY-SA 3.0. License URL: https://creativecommons.org/licenses/by-sa/3.0/deed (accessed on 22 July 2022).

The industrial revolution was already underway at this time. Many people had left the countryside, going to the city to earn a better living. Cities were often dirty and crowded. People were interested in leaving the city and seeing the beautiful countryside, but often, they did not know where to go. Cameras were still very primitive and only shot black-and-white pictures. However, the printing press made it possible to publish a travel guide that would tell people where to go. Lacking pictures, Wordsworth relied on poetry to convey the deep means that would attract people.

Wordsworth's travel guide to the Lake District includes surface meaning or information about how to get to certain places and the best order in which to visit them. However, his poetry gave him the ability to convey deep meaning about the landscape that made people want to come and visit. Wordsworth's best-known poem related to the Lake District was titled, "I Wondered Lonely as a Cloud." It is also often referred to as "Daffodils."

### I Wandered Lonely as a Cloud (The Daffodils)

By: William Wordsworth (1802) [22]
I wandered lonely as a cloud
That floats on high o'er vales and hills,
When all at once I saw a crowd,
A host, of golden daffodils;
Beside the lake, beneath the trees,
Fluttering and dancing in the breeze.

Continuous as the stars that shine
And twinkle on the milky way,
They stretched in never-ending line
Along the margin of a bay:
Ten thousand saw I at a glance,
Tossing their heads in sprightly dance.

The waves beside them danced; but they
Out-did the sparkling waves in glee:
A poet could not but be gay,
In such a jocund company:
I gazed—and gazed—but little thought

What wealth the show to me had brought:

For oft, when on my couch I lie
In vacant or in pensive mood,
They flash upon that inward eye
Which is the bliss of solitude;
And then my heart with pleasure fills,
And dances with the daffodils. By: William Wordsworth (1802) [22]

This poem is full of deep meaning. He refers to daffodils as the crowd. He says that they are fluttering and dancing in the breeze. This poem conveys the type of deep meaning that fascinates people and makes them want to visit the Lake District. It is this type of deep meaning that hopefully will motivate the people of Virginia to nominate and visit scenic viewsheds.

### 3. Seeing Deep Meaning in the Literature: An Application

The proposed Scenic Virginia Viewshed Program demonstrates how "deep meaning" can be included in scenic quality assessment. The entire scenic assessment procedure is described in a paper titled "Virginia scenic viewshed assessment project" [23] in the proceedings of the 2019 visual resource stewardship conference: Seeking a 2020 Vision for Landscape Futures [24]. How deep meaning is identified is described below. A grant was provided to Virginia Tech by Scenic Virginia to undertake a literature review and propose a viewshed assessment procedure.

*Looking for Deep Meaning in Virginia Landscapes: The Literature Review and Serendipity*

The literature review drew upon two databases. The first database was compiled from keyword searches by Virginia Tech researchers and contained 853 articles published between 1969 and 2018. The eight keywords used to search for journal articles and books were scenic value, scenic beauty, scenic quality, visual quality, visual resource management, visual assessment, landscape preference, and landscape quality. In addition to journal articles, the first database included abstracts and is searchable. The literature review database can be accessed online [22].

Virginia's landscapes are varied. The Virginia Tech research team needed to test some of the scenic assessment methods found in the literature on Virginia landscapes. However, there was no database of public preferences for Virginia landscapes. The researchers were stumped. Then, it occurred to them that they might use photographs from Scenic Virginia's Scenic Photo Contest to evaluate some assessment methods from the literature review. Could this set of photographs be a surrogate for a public preference survey? The citizens of the Commonwealth of Virginia would not have submitted their photos to the contest unless they believed the photos depicted scenic landscapes. Some photographs were eliminated because the owners used photographic techniques that did not realistically capture the visual characteristics of the landscape contained in the photo.

As the researchers examined the photos using different techniques and metrics from the literature review, they serendipitously found something missing. The techniques and theories from the literature review could not capture all aspects of some of the most scenic landscapes. It became clear that many contained content that had a deep meaning. Four categories of deep meaning that were part of the landscape but not adequately reflected in the scenic quality assessment literature were identified and will be examined. These are historical content, cultural content, urban content, and ephemeral content. It was a serendipitous discovery. These four categories and how they were used in the Virginia viewshed assessment process is described further below. Note: these content categories were assessed along with other more traditional scenic assessment variables and summed to provide a scenic assessment score.

## 4. Seeing Deep Meaning in Virginia Landscapes: The Exploring an Application of Deep Meaning in Scenic Quality Assessment

Scenic Virginia's proposed viewshed program provides an excellent example of the potential role "deep meaning" can play in scenic quality assessment. Unfortunately, there is not enough space here to describe the entire scenic viewshed assessment procedure [12]. Instead, the use of deep meaning to assess scenic quality will be described here.

Most visual assessment procedures identified in the literature review are for public, undeveloped lands. The underlying assumption is that a natural landscape is more scenic if it does not have human alterations. However, it was clear that some of the scenic contest photographs contained significant human alteration, and that the deep meaning of the content enhanced the scenic quality of the landscape depicted.

The proposed procedure for assessing Virginia viewsheds has two parts: nomination and assessment. The nominations will be made by citizens or local officials. The nomination will solicit two types of information: surface information and deep meaning. The surface information will be descriptive facts such as location and size of the viewshed. To obtain deep meaning, the nominator will be asked for a name and description of the viewshed that conveys its importance to local people. Supporting materials and examples are provided to the nominator in order to encourage the use of words and descriptions that convey deep meaning. Once the nomination is received, a scenic assessment will be conducted by someone knowledgeable about the proposed scenic assessment criteria to determine if the viewshed meets the criteria necessary to be placed on the proposed scenic viewshed registry. There are six scenic quality criteria that are rated on a three-point Likert scale. Four of the criteria assess traditional scenic quality measures found in the literature and two address criteria related to deep meaning, not found in the literature. The criteria ratings are summed to provide a scenic quality score. In addition to scenic quality an assessment if public concern is also conducted. The combined scores determine eligibility for placement on the scenic viewshed register. These measurement scales will need to be researched further in a pilot application to be sure they can be understood by those using them and that the results are reliable. Some landscapes also contain negative content. Negative content includes things in the landscape that are incongruent or distracting and have a negative effect on the viewing experience. Negative content is addressed in the literature and can involve deep meaning. However, since it is already in the literature it is not addressed further here.

The purpose of this analysis is to examine the identified four categories of content that can occur in the landscape to convey deep meaning. Historical content, cultural content, and urban content were rated as different types of "positive human influenced content in the viewshed" on a three-point scale: visually striking (2 points), noticeable but not striking (1 point) and not visible (0 points). Ephemeral content was rated on a three-point scale: frequent/predictable (2 points), not frequent but predictable (1 point) and not predictable (0 points). Again, these will need to be tested in a pilot study for consistency and validity. Each of the deep meaning content categories are described below with a photographic example.

### 4.1. Historical Content

Historical content (Figure 6) conveys deep meaning about the past that is no longer present. It is often a more nostalgic meaning. Landscapes placed on the National Register of Historic Places may convey historical content, but not always. Deep meaning from historical content should not be confused with historic significance. For example, a landscape may be placed on the National Register of Historic Places because an event important to our history took place there but is not visually evident. If a viewer does not sense an event from the past, there would be no deep meaning. Likewise, historical places may convey deep meaning about the past, but the past may not be significant enough to our history to merit being on the register of historic places. Deep meaning historical content conveys images of human use and activity that existed previously. It is a remnant of past activity.

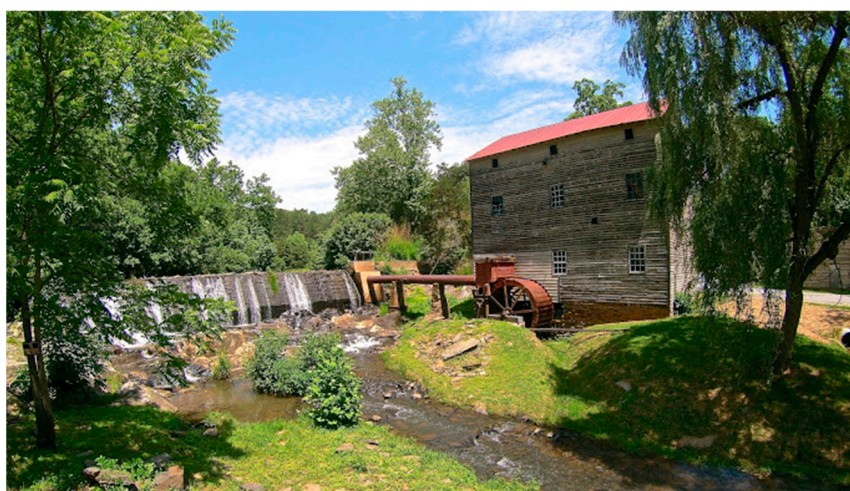

**Figure 6.** Historical content conveys deep meaning about past activities that have left a mark on the landscape. Activities that no longer exist may continue to bring images to mind. The ruins of the mill depicted in this photograph are reminiscent of the water wheel turning, the mill stone grinding, and sacks of grain being carried away to feed people and animals. Historical content does not have to be historic or connected to an important historic fact or event, although it could be. (Brightwells Mill by Bill Knarr Courtesy of Scenic Virginia).

*4.2. Cultural Content*

Cultural deep meaning results from landscape alterations by humans (Figure 7). Cultural deep meaning comes from mental images of human activity or settlement in the past, not historical meaning but human land use such as farming and ranching. The tobacco barns in Virginia or the lobster docks and boats on the Chesapeake Bay are examples of deep cultural continents. Cultural content conveys a sense of belonging. It does not feel out of place in the landscape. Cultural deep meaning is assessed by how visually evident it is. It must be visible to be rated.

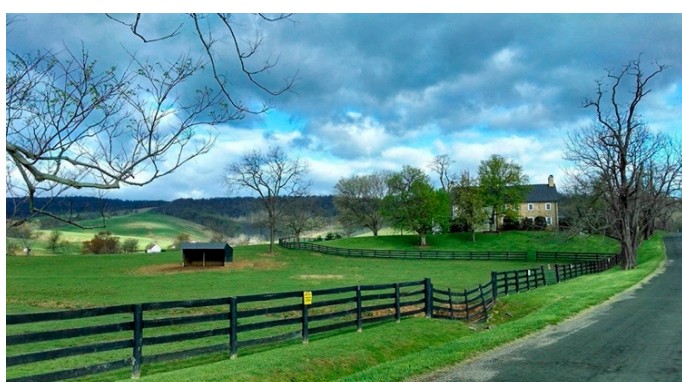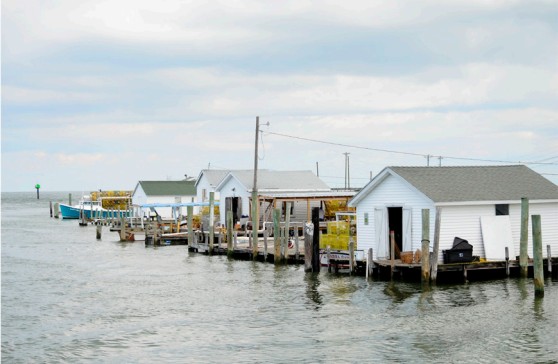

**Figure 7.** Cultural content conveys a deep meaning of human activity. The fences, pastures, and animal sheds in the image on the left convey deep meaning about the agricultural activity The fact that it appears well maintained conveys a feeling of cultivation and care of the land and animals. (Titled "April in Paris" by Janet White, courtesy of Scenic Virginia—photo has been altered.) The fishing huts, boats, and crab traps in the image on the right convey the deep meaning of fishermen and their boats going out to sea and bringing back crabs. (Titled "Huts in Tangier Island" by Arun Kumar, courtesy of Scenic Virginia.)

### 4.3. Urban Content

The urban content (Figure 8) is another type of cultural content. However, it is in a separate content category because urban environments can often be either positive or negative in the deep meaning they convey. Positive urban content is visually striking and conveys the deep meaning of dynamic and progressive human culture. Therefore, the urban content is an assessment of how visually striking the content is. Urban content must also be visible to be rated.

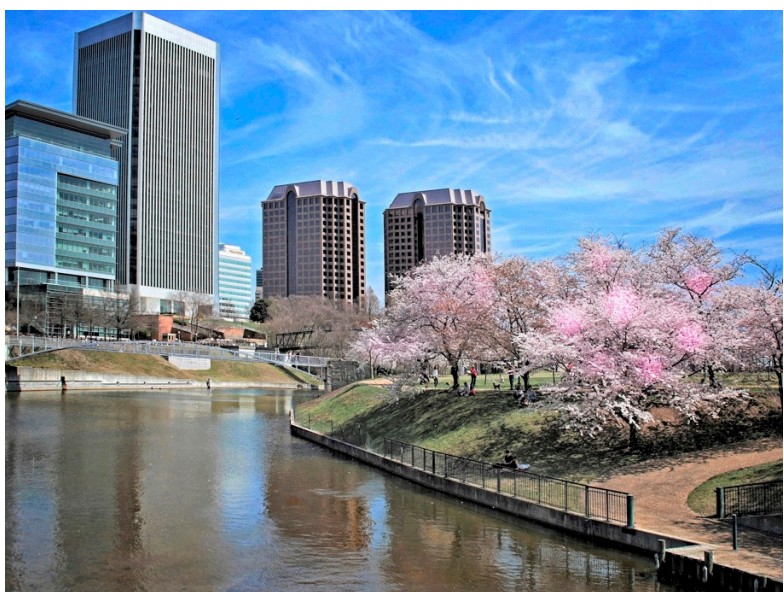

**Figure 8.** Urban landscapes can vary a lot in the deep meaning they convey, some of which may be negative deep meaning. Urban deep meaning often has an architectural component that conveys deep meaning about the activities that occur there. In the photo above, the juxtaposition of the water and green space with the office buildings conveys deep meaning of successful, progressive businesses as well as pleasant places to walk and escape the fast-paced business environment. (Titled "Cherry Blossom Time in Richmond" by Nancy Helms, courtesy of Scenic Virginia.)

### 4.4. Ephemeral Content

Ephemeral content in the landscape also contributes to deep meaning. Ephemeral content is visual content that changes at different times (Figure 9). How can ephemeral content contribute to scenic quality if it changes? First, it needs to be predictable and relatively frequent. For example, an erupting volcano may provide a scenic display. However, if it only erupts every thousand years and the timing is unpredictable, then it cannot contribute to scenic quality in a useful way. Conversely, ephemeral content that is predictable and occurs on a relatively frequent basis does contribute to scenic quality. For example, the color of fall leaves and migrating waterfowl occur annually and are predictable and reasonably frequent. Both convey deep meaning and contribute to the scenic quality of the landscape.

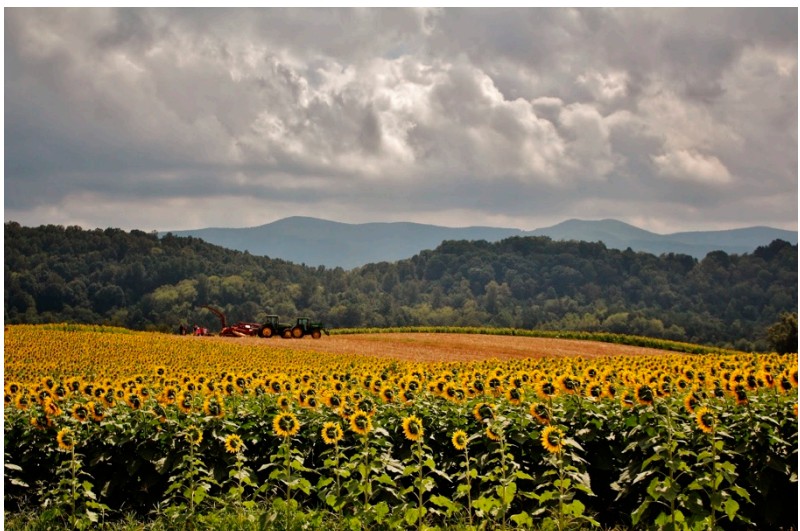

**Figure 9.** The sunflowers in this image contribute deep meaning about seasonality and the reproductive aspects of nature, as well as the human agricultural activities that produced this landscape. This landscape will occur reasonably often and is predictable. (Titled "Beaver Dam Farm" by Sandra Deans, courtesy of Scenic Virginia.)

## 5. Seeing around the Bend: Discussion

This essay argues there is a need for a scenic assessment approach that will be more meaningful to citizens who live near scenic landscapes that are worthy of scenic preservation. Much of the scenic assessments carried out for the last sixty years have been on large-scale public lands managed by the federal government. In this context, there has been little need for an assessment approach that would be more meaningful to local people. This has resulted in little development of new assessment methodologies for the last 30 years and less knowledge-sharing between disciplines. However, things are changing.

In Europe, the European Landscape Convention is changing how things are assessed. It is putting more emphasis on public engagement. As the United States moves towards more state and local scenic assessment, there will be need for public support and an assessment process that is more meaningful to the public.

This essay advocates the need for experiential understanding or deep meaning in scenic assessment that will result in a more meaningful process to people in a local area or region. It is clear from an examination of photographs from the Virginia Scenic Photo Contest that there are components in the landscape that contribute to scenic quality but would not have been be taken into account in the methodologies in the literature. These photographs are images that members of the public believed were scenic or they would not have submitted them to the contest. Four content categories that contain deep meaning and contribute to scenic quality were identified and were included in the proposed Virginia Scenic Viewshed Assessment process. Obviously, the proposed process needs to be tested to be sure that the process can be understood and that it can produce meaningful and consistent results. That work is already underway.

As visual management moves beyond protecting public lands and mitigating the visual impact of intrusive landscape alterations, we must look forward. We must see around the bend to ensure that the scenic quality assessment tools needed will be helpful and available. We must develop and use new tools that recognize deep meaning as a criterion that is important to the sense of a place where people live.

**Funding:** This research received no external funding.

**Data Availability Statement:** Not applicable.

**Acknowledgments:** This essay came out of the research on the Virginia Viewshed Study completed by faculty and graduate students at Virginia Tech. The research was funded by a grant from Scenic Virginia. The author is indebted to the following individuals who worked on and reviewed the Virginia Viewshed Study: Lynn Crump, PLA, ASLA, Scenic Virginia Viewshed Project; Richard Gibbons FASLA, Landscape Architect (retired) DCR-Planning & Recreation Resources Commonwealth of Virginia; Leighton Powell, Scenic Virginia, Sponsor of the Virginia Tech Viewshed Research Project, provided feedback and editorial review for Virginia Tech Viewshed Research Project; and Jisoo Sim, Virginia Tech, worked on the Viewshed Assessment Research Project and is at the Korea Research Institute for Human Settlements.

**Conflicts of Interest:** The author declares no conflict of interest.

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
