# Peer review of "Deep Meaning in Scenic Assessment: Seeing around the Bend"

_land, doi:10.3390/land11101646_

Round 1

Reviewer 1 Report

Thank you to the author for an interesting contribution to the methodologies currently used in the scenic [beauty] assessment of landscapes. The incorporation of 'deep meaning' values in scenic beauty assessment of landscapes, although debatable due to its implementation complexity (and subjectivity, some might add), is worth pursuing. 

As general comments, I suggest the author to consider adding information on the following three issues:

1. As means of widening the discussion to a broader geographic context, it could be useful to mention the European Landscape Convention in the introduction section. This is a relevant discussion since the ELC aims to increase awareness among the civil society, private organizations, and public authorities of the value of landscapes, involving all stakeholders in the process of identifying landscapes (i.e. delimiting and describing landscape units, considering people's perception), as well as implementing landscape policies aimed at [all] landscapes protection, management and planning;

2. It would also be interesting to mention other authors that defend that personal experience is a factor to be considered in the aesthetic response of humans when contemplating landscapes. Two authors (and specific publications) come to my mind on this subject: 

Riley, Robert B. 1992. ‘Attachment to the Ordinary Landscape’. In Place Attachment, edited by Irwin Altman and Setha M. Low, 13–35. Human Behavior and Environment. Boston, MA: Springer US. https://doi.org/10.1007/978-1-4684-8753-4_2.

Ulrich, R.S. 1983. ‘Chapter 3 - Aesthetic and Affective Response to Natural Environment’. In Behaviour and the Natural Environment, edited by I. Altman and J. Wohwill, 6:39–83. Human Behaviour and Environment. New York: Plenum Press.

Other, 'newer', authors/publications are very welcome to this discussion

3. Conclusions (section 5, in my view) should be improved and made clear that, although many consider the incorporation of 'deep meaning' values in scenic assessment methodologies to be difficult (due to the complexity and subjectivity involved), the present proposal does take steps forward on this matter, proposing a method that is operational, repeatable and based on objective criteria.

Specific comments/corrections proposed:

4. (lines 35-36) "Many of the scenic quality tools we have today were initially developed to preserve the visual quality of public lands...": please add citation to support this statement (e.g. Daniel & Boster (1976) - The SBE Method);

5. (line 59) "This period is often referred to as the Environmental Movement": Please add citation (e.g. Daniel, T. C., and J. Vinning. 1983. ‘Chapter 2 - Methodological Issues in the Assessment of Landscape Quality’. In Behaviour and the Natural Environment, edited by I. Altman and J. Wohwill, 6:39–83. Human Behaviour and Environment. New York: Plenum Press.);

6. (lines 140-143): some sentences repeat what was said in paragraph above Figure 2 - please remove repetitions;

7. (line 157): "... had a deeper meaning for each rock..." - please correct typo;

8. (Figures 2 and 3): is it possible to improve images? Make them closer to label, if possible. Also, it is not clear if image in Figure 3 is a good of a bad example of 'deep meaning' in a wine bottle label. In my perspective it is a bad example (starting with the name of the wine), but the figure's caption seems to point in the opposite direction. Maybe this point can be made clearer;

9. (lines 229-232): again, remove repeated sentences from text above Figure 5;

10. (line 278): "How deep meanings were..." - please correct typo;

11. (line 288): "The literature review can be accessed online [12]." - this web location can only be accessed by asking permission, please try to grant [immediate] access to anyone using the link;

12. (lines 303-304): "The four contents are: historical content, cultural content, urban content, and ephemeral content.- please correct typos (also in sections 4.4 and 4.5: 'content', not 'continent');

13. (lines 362-367): what about negative 'deep meanings'? How can they be rated?

14. (lines 382-384): "Conversely, ephemeral continent that is predictable and occurs on a relatively frequent basis does contribute to scenic quality." - substitute 'However' by 'Conversely' (and check if meaning of statement hasn't changed);

Final observation: apart from the publications cited from the 2019 Visual Resource Stewardship Conference (and reference [1]), all references are from the 1960's to the 1980's decades. We would advise the author to try to include references of recent publications in order to underline the relevance of the present proposal in today's 'state-of-the-art' discussion on scenic beauty assessments and sustain its interest in the current debate (within this area of knowledge). 

Again, thank you for the interesting reading provided

Author Response

Thank you. Your comments were very helpful.  I agreed with all of them and responded accordingly.  See attachment.

Reviewer 2 Report

Paper is very interesting for readers and it gives insight into very important issue of the landscape visual assesment. I would like suggest to expand paragraph 2.1. Seeing back. Context with following works which refer to the dual character of the landscape:

Ipsen, D. (2012). Space, place and perception: The sociology of landscape. Exploring the boundaries of landscape architecture. S. Bell, I. Sarlöv Herlin, & R. Stiles. 60–82. Abingdon: Routledge. 

Hale, J. (2016). Merleau-Ponty for Architects (1st ed.). Routledge. https://doi.org/10.4324/9781315645438

Author Response

See attached response.

Reviewer 3 Report

The visual values of the landscape are a very rich topic. This manuscript takes a poetic perspective toward the valuation of deep meanings that are contained within landscapes. It is a very interesting approach. However, many aspects should be improved. 

Firstly, the background (state of knowledge) presentation should include more references. When taking up such a topic, even in the form of an essay, one cannot ignore such important theories as iconography and iconology or semantic-based approach to landscape. World literature contains many other concepts that refer to landscape perception and could be mentioned, e.g., phenomenology, landscape perception studies, Gestalt...

Secondly, the methodology concerning the analysis of photographs is not adequately described. We know neither the number of photos nor the number of experts. And what did they do with the pictures exactly? Details of the procedure are not sufficiently described. We only learned from the results that there were three different levels of rating. The methodology must be explained clearly and strictly. I suggest adding a table or a diagram to explain the procedure in a simple way. Other than this, the link to the literature review has restricted access.

The results should be presented in more detail. Every subsection of the results states that the landscape content was rated on a three-point scale. But we still don't know the final effect. What was the purpose of the rating? And what did it bring in the end? Did most of the scenic views contain visible historical or ephemeral content, for example?

The discussion part should present the results against the background of the existing state of knowledge. The goal of this part is to highlight the novelty of your research. What new things were ascertained? 

Last but not least, the conclusions must stem strictly from the research presented in the article.

Minor remarks include:

- lines 132-134 and lines 141-143 are a repetition

- lines 303-304 - is this correct: "historical continent, cultural continent, urban continent, and ephemeral contact"?

 - the words "content", "contend", "contact", and "continent" seem to be mismatched. Most likely, an automatic correction engine was used, and it replaced repetitions with similar words, which changed the meaning. Use the search tool to find these words and double-check whether they were used consistently.

Author Response

Reviewer 3- 

Thank you for your comments.  I have responded to them in a separate attachment.

Round 2

Reviewer 1 Report

Thank you for the changes and additions introduced in the manuscript. All major issues were clarified and the references added are quite relevant and adequate.

Please revise the list of references - it seems references 16 and 17 are the same. Also, in reference 12, the part before the author's name ("Page 1336 -") should be eliminated.

I am pleased to recommend the publication of the manuscript as is, after a brief style check.

Author Response

Dear Reviewer 1, thank you for your recommendations.  I agree with your recommendations and have made the recommended changes.

Reviewer 1 comment:  Please revise the list of references - it seems references 16 and 17 are the same. Also, in reference 12, the part before the author's name ("Page 1336 -") should be eliminated.

Author response:  I agree with the comment and have made the change as indicated.

Reviewer 1 comment:  I am pleased to recommend the publication of the manuscript as is, after a brief style check.

Author response:  I have reviewed the manuscript and made the final edits.  I asked another person who is good at proofreading also to review the manuscript.  I have made changes that can be seen in Microsoft Correct changes.

Reviewer 3 Report

With the modifications made, the nature of the manuscript, its goal and details of the procedure are much clearer to the reader. The manuscript still needs proofreading to eliminate small errors, e.g., double spaces.

Author Response

Author comment:  Thank you Reviewer 3 for your comments.  I am in agreement with your comments and have done as you recommended (see below).

Reviewer 3 comment:  With the modifications made, the nature of the manuscript, its goal, and details of the procedure are much clearer to the reader. The manuscript still needs proofreading to eliminate small errors, e.g., double spaces.

Author response:  I have proofread the manuscript again and corrected many small errors.  I have asked another person who is very good at proofreading to proofread the manuscript.  Our combined changes can be viewed on the manuscript with track changes turned on.   

Thank you.